# Understanding Protective Factors for Men at Risk of Suicide Using the CHIME Framework: The Primacy of Relational Connectedness

**DOI:** 10.3390/ijerph20032259

**Published:** 2023-01-27

**Authors:** Katherine M. Boydell, Alexandra Nicolopoulos, Diane Macdonald, Stephanie Habak, Helen Christensen

**Affiliations:** 1Black Dog Institute, Sydney 2034, Australia; 2Faculty of Medicine, University of New South Wales, Sydney 2034, Australia; 3Dalla Lana School of Public Health, Faculty of Medicine, University of Toronto, Toronto, ON M5G 1V7, Canada

**Keywords:** male suicide, protective factors, CHIME recovery framework, qualitative research, relational connectedness

## Abstract

Suicide is a global problem, ranking among the leading causes of death in many countries across the world. Most people who die by suicide are “under the radar”, having never seen a mental health professional or been diagnosed with a mental illness. This article describes the protective factors for men experiencing suicidal thoughts, plans, and/or attempts who are “under the radar”. Using in-depth qualitative interviews, we aimed to understand stakeholder perspectives on the protective factors that influence men’s wellbeing. The pervasiveness of relational connectedness in men’s narratives was identified as a central protective factor. Other key protective factors included meaningful activity, empowerment, and hope. These results have the potential to facilitate the development of focused community initiatives. More generally, the current research offers an example of a qualitative inquiry into men’s wellbeing that focuses on strengths and positive factors in their lives and may provide a guide for future community-based suicide prevention research.

## 1. Introduction

Suicide is a global problem, ranking among the leading causes of death in many countries across the world [1,2] The Australian Institute of Health and Welfare indicates that approximately 3000 people die by suicide each year in Australia, with suicide being the leading cause of death for people aged 15 to 44. There is a glaring gender imbalance in global suicide rates, with men representing close to 80% of all suicide deaths [1,3,4], making suicide the leading cause of male death. 

Men can be reluctant to seek help for suicidality [5]. Han and her colleagues’ review identified factors influencing this lack of help-seeking, including high self-reliance, the lack of perceived need for treatment, and stigmatizing attitudes toward suicide, mental health challenges, and professional treatment. There is great diversity in men’s help-seeking practices, given that some men do not adopt dominant masculine discourses that thwart help seeking and take up other caring discourses that support help-seeking [6]. 

Traditional or mainstream suicide prevention is grounded in a risk paradigm, yet individuals represent more than risk factors. It is important to identify protective factors for at-risk individuals. There is a need to question the status quo in understanding and preventing suicide. Critical suicidology is a paradigm that offers a more subjective, historical, ecological, social justice-oriented, poetic, and socio-political approach than customary suicidology studies. This more contextualized approach allows suicide prevention efforts to capitalize on individual and community strengths. The goal of this paper is to present the protective factors for men experiencing suicidal thoughts, plans, and/or attempts and who identified as being “under the radar,” that is, reported that they had not sought help from the formal health system in the past year. The research team discovered during the interviews that many men had in fact been in touch with formal services in the past year, yet still view themselves as under the radar [7]. To our knowledge, there are no published studies examining suicide protective factors among men considered under the radar. The data corpus for analysis was based on qualitative interviews that were collected as part of the “Under the Radar” study, a mixed-methods project exploring the views, attitudes, and characteristics of men who have experienced suicidal ideation, plans, and/or attempts but are not in touch with formal mental health services, as well as their preferences for support. 

Protective factors can be defined as societal or psychosocial circumstances or individual behaviors that lessen the likelihood that an individual will engage in suicidal behavior [8]. The study of resilience—the ability of individuals and systems to manage effectively in the face of significant hardship or suicide risk—has been posited as a valuable way of recognizing protective factors [8]. Although the capacity for resilience develops and changes over time, it is heightened by protective factors within the individual, the system, and the environment and contributes to the maintenance or enhancement of health. Masten and Powell [9] note that resilience requires “(1) that a person is ‘doing okay’, and (2) that there is now or has been significant risk or adversity to overcome.” Resilience can be further broken down into two forms: incidental and reactive. Incidental resilience represents something that an individual has been engaged in for some time that promotes health and wellbeing and becomes a critical component of coping when hard times arise. Reactive resilience is something that promotes health and wellbeing that an individual engages in as a direct response to difficult situational circumstances.

## 2. Materials and Methods

### 2.1. Theoretical Underpinnings

We align ourselves with critical suicide studies, which pursue socially-just, democratic, and ethical means by which to integrate alternative views and ways of being. This comprises the inclusion of the voices of lived experience and other methods that are “made discursively inaccessible via institutionalised knowledge practices” [10]. In critical suicide studies, thinking about people in terms of their risks is disrupted; they are viewed as much more than their risk factors [11]. 

### 2.2. Aims and Study Design

This phase of the Under the Radar project used qualitative methodology to collect data through semi-structured, in-depth interviews. Through its exploration of lived experience in its natural environment, qualitative inquiry can contribute to a broader understanding of knowledge [12]. The aim of this study was to explore how narratives of suicide were articulated by men who identified as being “under the radar” as well as loved ones bereaved by suicide. Online, in-depth interviews were conducted with 37 men who had experienced suicidal thoughts or behaviors. The interview team possessed both research skills and experiential wisdom.

### 2.3. Participant Recruitment

Participants were recruited through social media advertisements [Facebook, Twitter, Instagram, and LinkedIn] and external partners and organizations. Recruitment took place from August to October 2021. Participants were required to be 18 years of age or older, living in Australia, comfortable with the interview being conducted in English, and to have (i) identified as currently or previously being a male “under the radar” (not in contact with formal services in the past year). Ethics approval was received by the administering institution.

### 2.4. Data Collection

A co-design approach with people who have identified lived experience of suicidal thoughts and/or behaviors was used to develop the guideline questions that formed the semi-structured, in-depth interview (see Appendix A for the guideline questions). Co-design between the research team and lived experience advisors also contributed to developing prompts throughout the interview that were aligned with the research questions and goals. 

### 2.5. Data Analysis

Data analysis focused on the entire dataset of 37 men. Reflexive thematic analysis (RTA) allowed the team to take an organic approach to analysis, permitting a comprehensive and theoretically adaptable application of Braun and Clarke’s analytic approach [13]. This approach facilitated the identification and analysis of patterns or themes in the dataset. RTA is independent of a specific theoretical framework, allowing for broad and flexible application of the analytic approach across a range of epistemologies [14]. 

The six phases of reflexive thematic analysis were followed: familiarizing with the data, generating initial codes, generating themes, defining and naming themes, and writing up a report. Although these phases imply linearity, our analysis was recursive and iterative, and we moved back and forth between the phases as required [13]. We put reflexivity to work by reading the data for “gestalt” and engaging in generative coding [15]. For this paper, an additional step was taken to match the narrative text to the CHIME components in a deductive manner. Adding a deductive or concept-driven approach allowed us to apply the CHIME conceptual framework for personal recovery to the qualitative data corpus. This framework, created by Leamy et al. [16], followed a systematic review of 87 articles on frameworks used for personal recovery in mental health. It is the most wide-ranging portrayal of the recovery process to date and is known for its adaptability. Yung et al. [17] noted its application in studies as widespread as those on cultural diversity and depression [18] and art therapy for mental health recovery [19]. The framework consists of five domains: (1) connectedness, (2) hope and optimism, (3) identity, (4) meaning in life, and (5) empowerment. 

Subsequent to perusing the entire dataset, codes were then allocated to the appropriate sub-domains of CHIME. Rigor was addressed by applying well-established trustworthiness criteria via prolonged engagement with the subject matter, persistent observation of experiences, and perspectives about suicide and researcher triangulation. Team discussion in routine analysis meetings included a collaborative and reflexive approach, wherein we endeavored to achieve richer interpretations of meaning.

## 3. Results

### 3.1. Participant Characteristics

The demographic characteristics of the participants are described in Table 1. The average age of participants was 47.5 years, with almost half reporting that they were married. The vast majority identified as heterosexual (86%) and living with others (73%). With respect to the diagnosis of the men in our study, we did not assume that mental ill-health was present—in fact, many of the men in our study explicitly talked about the fact that it was not mental illness that underpinned their suicidal thoughts or behaviors, but rather other social determinants such as relationships, financial status, workplace situation, and previous childhood trauma.

### 3.2. CHIME Domains

Of the five CHIME domains, connectedness and meaning in life mapped onto the data most frequently (Figure 1). The remaining domains of hope and empowerment were less frequent, with identity found to be present in the data far less frequently. We note that connectedness and interpersonal relations are currently posited as one component of the framework and agree with Price-Robertson et al. [20] that this domain is better conceptualized as permeating all of the domains, including experiences such as hope and optimism, identity, and empowerment. The domains other than connectedness have been noted as purely intra-psychic processes or achievements [20]. The very structure of the CHIME framework, where connectedness sits aside the four intra-personal recovery processes (hope, identity, etc.), reinforces the view that what goes on inside people’s minds is of a vitally different order from what occurs in their social interactions. As noted by Price-Robertson and colleagues, “Experiences such as hope, meaningfulness and empowerment emerge at the intersections between people, their relationships and environments; they are best seen as interactional processes rather than states possessed by any one individual.” (p. 5). 

#### 3.2.1. Connectedness

Connectedness refers to individual relations and wider community and societal connections. Five subthemes were identified that related to a sense of connectedness, which operated as a protective factor in men’s lives. They centered around having someone to live for, not wanting to cause others pain, experiencing unwavering support from others, a positive experience of formal services and supports, and the positive role of peers.

Family Members to Live for

For the men who identified as having others to live for, most were referring to their children. They worried that their suicide would be extremely upsetting to others, particularly those who relied on them both instrumentally and emotionally. As the following quotes indicate, men were very explicit about the fact that thinking of their children and the impact suicide would have on them was a protective feature.

*I’ve got two kids and that is always the thing that stands in the way and that, yeah, it’s not something I could do to them…But I would say it at the heart … [I] always talk myself out of it - is the kids…I am really close to both of my kids, so it’s like we are fairly open about. So yeah, so I’ve certainly thought of ways that I would, um, kill myself, so if I didn’t have that to stop me…didn’t have them stop me*.[Dan]

*I couldn’t put my kids through losing another parent*.[Connor]

Not Wanting to Cause Pain to Others

Respondents often identified that the one factor that unfailingly caused them to pause and prevented them from pursuing suicide were their thoughts of the pain and suffering their deaths would cause other people. The broader impact and ripple effect were considered by many men in our study.

*You know it [suicide] doesn’t solve the problem. It leaves it to somebody else*. [Ross]

*I know how much it would devastate them, and that’s the last thing in the world I want to do. I’d like to do it, but I know if I do it will affect other people… it doesn’t solve any problems. It just transfers them so somebody else then have to deal with that and those affected*. [Matthew]

*I feel I could never realistically ever take my life because of my dad and my brother. I can’t complete that cycle, you know, I and I can’t do that to my sisters’ account. Do that to my greater family who have been through it twice before*. [Stefan]

*If you do this, it’s gonna not just impact your immediate family, it’s going to impact the the wider… the much more extended family and beyond that community, you know*. [John]

Unwavering Support from Social Network

Many men identified the unconditional support and understanding that they received from significant others in their network that provided the motivation for them to keep going. Significant others often included a partner, but men also identified the role of their mates in the workplace, which is also highlighted in the meaning men identified in workplace relations. 

*So well, she’s a huge part of my life and she’s put up with a lot of my anxiety and depression for many, many years. Just been incredible support, so without [her] you know I’m I may not be here, I don’t know*.[Tom]

*My partner helps talk me down*. [Justin]

*I’ve got a very supportive partner. Most of my friends are, like myself, military veterans so, a fairly strong understanding of trauma and suicide and things like that and, for a group that historically never used to talk about those… they’re [partner] probably more helpful than the most circles of friends, I guess because they all know somebody who has been there, done that*. [Ross]

*I had some good people around me which was really helpful, a couple of people just in [work] my team here. We could, you know, we could quite easily just turn off work and just go for a coffee and talk about stuff you know*. [Darren]

Helping Professionals

Although men identified as being “under the radar,” which was defined by the research team as not having experienced formal services in the past year, many men who were recruited revealed during the interview that they had used services in the past year. Those who had not used formal services recently drew upon previous positive experiences with helping professionals in the health sector. 

*I certainly think that my psychoanalysis helped me*. [Joel]

*I went on to see that GP and actually [he’s] been amazing. Uh, so he’s sort of acting like a psychiatrist via proxy, like he’s doing all the hard work and you know he’s willing to challenge everything and he certainly makes himself available and his language is open and free*. [Stefan]

*And talk through all this over a long time with this counsellor and he was brilliant*. [Jacob]

*My psychologist was largely, relatively available when needed. So, I was able to have regular sessions but also show there were times when I was in crisis. She was able to fit me in within a couple of days. And she bulk-billed*. [Justin]

Several men’s narratives included their reflections on the beneficial ways in which service providers enabled them to reframe their situation and advised them to identify their strengths and devise coping strategies for dealing with their distress.

*Regarding psychoanalysis as a waste of time, and I don’t see it like that, but I found it quite beneficial. And it was actually through that process and I actually began to recognize a number of my positive and negative strategies of how I come to cope with my experiences as a child*. [Joel]

*Having a psychiatrist, right, and understanding more about my diagnosis and the reason for... some of the actions and outcomes and pinning them back to a diagnosis allows you to it’s not an excuse, it’s an explanation*. [Mitch]

Connectedness with Peers

Respondents repeatedly identified the importance of regular social interaction with others. Some discussed the significance of feeling socially connected without necessarily talking about their daily struggles and challenges. 

*Close friend of mine who always invite me to walk his dogs and a dinner with his family...So, I just feel it’s very nice to have a companion that’s actually truthful to you*. [Tyler]

*For hours and forget about...troubles. It was, yeah, that social interaction with the other boys at the gym*. [Nate]


*Sit down and have a coffee at the local shopping centre*
[Justin]

*I know a lot of people, but I’m talking about friends, people I can really rely on and I can share stuff with and that you know they’re supportive of me. And you know, they’ll, you know, they’ll tell me where they think I’m stuffing up, so I’d also appreciate that*. [Jacob]

Several men identified the ways in which conversations with peers who had experienced similar stresses or challenges could be very helpful. Being with someone who had comparable or shared experiences was viewed as beneficial and resulted in a sense of feeling that one was not alone. 

*A psychologist is like half of the solution. You know the other one is people. You really need people around you. And I’ve got a few mates now that we have a little joke ’cause we call it the Broken Hearts Club. So that is a big part of it. It’s not family. It’s not close friends, because if people haven’t walked down that road, you know, it just means nothing. Blokes in particular don’t want to talk to someone who has no idea what they’re talking about. The only way I, and this is my experience, can feel like I can open up to people is when I know people that have been down that road*. [Alec]

*The Mankind project is... it’s a voluntary thing, it’s a worldwide organization… It’s been a positive thing in my life. all they do all kinds of training weekends you know around different things, basically bettering yourself. But they also have like what they call, an ‘I group’ usually once a week that you can go to, sit down in a confidential circle and basically making sure that you are accountable... genuine? … I’m still friends with a number of people from the I group. Like some still friends with helped me a lot over the years*. [Warren]


*Sitting in a group of men having a coffee. You know, saying, uh, you know, if you’re feeling you know that life’s crap, you know it would be…Catching up meeting with ten strangers to have an informal, social things to talk about. Life or not life and just say how are you coping with life at the moment or something and then hanging with other people going oh you know and I think that then gets you out of this echo chamber, which would be and that’s why I didn’t want to go to somebody and more much more formal thing… I suppose that might that have happened in the real world I might have had a bit more contact with friends and then even though we wouldn’t have discussed this, you would have sat there kind of listening to their problems and other things and go OK you know. And I suppose then that gets you looking at not being so insular and everything else and then. And I suppose you become very self very centric. How’s this on me on me on me versus wait a minute? Everybody else has these problems...a reality check*
[Edgar]

#### 3.2.2. Meaning

One of the five CHIME domains is centered around meaning—defined as a meaningful life and social goals and a meaningful life and social roles. Four subthemes were identified that aligned with meaning making: physical and other meaningful activities and being in nature; meaning in the workplace; pet ownership; and faith. The overlap between meaning and connectedness, whether via connectedness with the environment, pets, and/or people. 

Physical Activity and Being in Nature

Many men acknowledged the importance of fitness and physical activity in their lives and made the explicit connection between that and their overall sense of wellbeing. It offered an outlet for many; for some, it was an “escape”; and for others, the social aspects of physical activity represented an informal source of help and support.

*Yeah, that’s been my silver lining and I’ve you know through that I’m more active so I’ve got my fitness back. I’ve got healthy, I’ve got fitter or and you know by all that physical activity that’s good that’s allowed me to then you know, be more consistent with gym routines*. [Alec]


*Exercise was something that really helped me. Obviously. Yeah, mountaineering and hiking is something that I really like to do, and that was sort of my escape*
[Leo]

*I took up running and I find that running is very good for my mental health and I’ve certainly read quite a number of books now with regards to that. Uhm, and it helps, but it doesn’t, you know, completely help*. [Chris]

*I was involved in a couple of sporting group sporting clubs which helped get my physical and anger out. I was very active in the AFL in going to the footy at least every second week if not more*. [Adam]

*If I’m out in some open space, I feel like comfortable and safe, and so that’s just a place that I go*. [Stefan]

Some men also realized what would have been helpful, even if they themselves had not experienced it, as illustrated in the quote below:

*If I’d been part of a group or some form of socializing activity that would have been stopped me becoming isolated. Going out, even if it’s for a walk or ride or talking with people, even if it’s the guy in the shop, it just opens your mind and I’ve felt that maybe if that’s what was really important to be able to instil this need for social interaction. Um, um, no matter how minor it is. I think that that’s what was missing*. [Edgar]

Positive Work Environment

The work environment represented an important source of meaning for many respondents. Several men stated that their work environment was the inspiration that kept them going. It was the people within the work environment—the relational connections—that were identified as being meaningful. 

*I’ve since gone on to get a new job and, uh, heaps better company with really good people, back in a job that I actually liked doing and I often say to people that’s been the silver lining on this cloud is that I’ve landed in a job that pays me twice as much money. I actually like the job*. [Alec]

*But I think I’ve got the best job in the whole world. Perhaps that has kept me going. Having a job with you guys, so I actually have the best job in the whole world. I love my job*. [Abe]

*I got work in the mines and ended up buying a house over there and worked on a ship station and, and a lot of good times and met a lot of good people and sort of turned things around a bit for me and I started coming good*. [Connor]

*I had some good people around me which was really helpful. A couple of people just in my team here. We could, you know, we could quite easily just turn off work and just go for a coffee and talk about stuff you know. So it was pretty good. It was. But yeah, there were, there were good people around, yeah, but you know, I’d learned over the years to be a bit picky and choosy about who you share with and what you share ’cause not everyone gets it*. [Darren]

Pet Ownership

A sense of meaning, purpose, and connectedness with their pets was often mentioned by participants. 

*This and this might sound silly, maybe not, but the two little doggies are amazing. You know they, like, they ground you in a funny sort of way as well*. [Darren]

*I really have a little Frenchie, so she is pretty much everything to me*. [Larry]

The following quote describes the cat that this respondent explicitly stated was the reason that he did not end his life. He expressed concerns about what might happen after his cat died. 

*Probably sounds a bit silly, but yeah, just my cat. She’s been my support animal since I was two I think*. [Luke]

The overlap of dog ownership with physical exercise was also highlighted, as seen in the following quotes. These participants linked walking their dogs with taking time out for themselves, which helped ease the pain they were experiencing.

*To overcome that pain and those thoughts, hmm, I take my dogs for a walk every morning. Like it’s just my 5 minutes of my time*. [Tony]

*We’ve got a dog and I take the dog. And I’ve done it with her because he doesn’t command a thing. So, we go for a bit of a walk and then we’ll sit down. Just sit down...she didn’t want to go for a walk anyway, so she’ll just come and sit next to me and we’ll sit. But I find that’s quite helpful*. [Chris]

Faith/Religion

For a couple of men, their faith was a source of comfort to them, and it helped them to reframe and/or accept what was going on in their lives.

*Do you know there’s a big family and they’re not gonna let you crash? And similarly on the faith side of things, you know you, you’ve got that faith ’cause you believe that you know the God you believe in isn’t gonna let you crash over*. [Darren]

*The value in my faith and very much around the sort of other orientation, and not to disregard my own wellbeing, but not to get fixed and not to just get fixated on me. And to see, you know, other people come and how other people are struggling particularly, and also to accept that phase about life is struggle, which a lot of non-Buddhists don’t understand because they just sort of thinking. Well, you know, it sounds a bit weird. It’s not to say that you seek it out, but you just accept the bad things do happen. The whole idea of Buddhism is to reach enlightenment through meditation. In India, like the power, meditation is incredible*. [Nate]

#### 3.2.3. Empowerment

Empowerment as described in the CHIME framework involves a sense of personal responsibility, control over one’s life, and a focus on strengths. We would also add to this description to include other elements of empowerment that were identified in men’s narratives, such as a sense of self-reliance, inner strength, and agency. Self-help and embracing coping strategies, and self-soothing methods contributed to a sense of empowerment.

*My first port of call is to be able to monitor where I’m going. So I think that’s been helpful in keeping me away, if you like, from anything approaching a suicide attempt up until, I suppose recently. And so that that’s self-agency… I think just my own sort of self-awareness is really important and not turning it into a crisis myself, you know. So that we don’t, I think that to be careful, you know I’m all for men reaching out and getting help but we don’t want to do that at the expense of saying, you know, don’t sort of trash talk self-reliance. That’s probably what I’m getting at because I think that still remains a really important thing and our capacity to be able to handle stuff ourselves and so, you know, I’ve developed those skills over the years*. [Jacob]

*You had enough inner strength to get yourself through it*. [Tyler]

*Self-reliance, that’s probably what I’m getting at, ’cause I think that that still remains a really important thing and our capacity to be able to handle stuff ourselves and so you know I’ve developed those skills over the years*. [Jacob]

It was often meaningful activity that led to men feeling that they had a sense of control, as can be seen in the quote below, where Mark also talked about the importance of music in his life.

*I can use music to kind of change my mood. I’m sorry if I might be like you know, I might come home in a bit of a really upset angry mood or whatever and I can put some music on that will flip me over, so they’ll have like associations to different things and so you can like take my mind to a different place*. [Mark]

*I just keep myself busy. I threw myself into my music*. [Oliver]

*Sometimes it’s about reading, so sometimes the things that helped me in strange bizarre ways, actually reading about this stuff. And I guess it just helps for me to reinforce what I’m going through and like. It normalizes what I’m doing or what’s happening to me. So that’s another strategy, so there’s a couple of books*. [Mark]

#### 3.2.4. Hope

In the CHIME model, the hope and optimism domain refers to a belief in recovery, the motivation to change, having relationships that promote and inspire hope, thinking positively and valuing success, and having dreams and aspirations. A sense of future thinking was also related to the hope domain. Some men in our study did aspire to have a positive future filled with hope. This often involved being present and valuing the “here and now”.

*Realize that there is hope, there is help and there are so many supports. It’s just having the knowledge to tap into it or having the courage to use knowledge to tap into the network of support so you know, first and foremost, that’s exactly how I would, you know, I was giving myself advice*. [John]

*I hope that people have a purpose for living, and I suppose that gives me a purpose to live*. [Abe]

*I hope that I am a role model to the other people that buy me putting myself out there and. Yeah, I guess becoming a bit vulnerable, and, inadvertently, that has an effect on others. To be a bit of a role man, I never thought about until then, but I hope that it is that way*. [Leo]

## 4. Discussion

The application of the broad and encompassing CHIME recovery framework domains (connectedness, hope and optimism, identity, meaning, and empowerment) recovery model [16] was extended to men at risk for suicide who are “under the radar”. Applying the CHIME framework to a qualitative analysis of protective factors in preventing suicide reveals the many ways in which every day experiences, both within and beyond the formal health care system, shape personal processes that serve to protect against suicide. To our knowledge, it is one of the first studies to apply the CHIME framework to the suicide prevention field. Future research in this realm could add to this work and include a case control study using the CHIME framework or another study applying CHIME to a population of men who had never been suicidal.

CHIME could also be used to examine suicides by women and other genders/sexual orientations. Jenkin et al.’s modification of the CHIME framework, which added an “S” for safety, could be applied as well [21].

The sense of connectedness was identified in the transcripts as a protective factor, and this sense of connectedness was the most predominate of the CHIME domains in the narratives of our participants. The lives of individuals cannot be separated from the social contexts in which they are rooted; thus, we would argue that this connectedness permeates all domains of the CHIME framework. Personal experiences of meaning, hope, empowerment, and sense of identity are inseparable from the social spaces in which they are embedded and aligned with the connectedness domain. That is, people are relational beings, and psychological phenomena such as hope and empowerment are viewed as emerging from the contexts of relationships. The pervasiveness of relational connectedness in men’s narratives as a central protective factor is consistent with Fincham and Scourfield’s [22] findings in their work on sociological autopsies. Their findings, based on the examination of individual cases of suicide studied post-mortem, focused on a typology of gendered suicide where relationship breakdown was the principal trigger for males. 

The protective nature of families when there is an extant positive family dynamic has also been reported previously in suicide studies of specific target populations; for example, adolescents [23] and trans adults [24]. McLean et al. [8] showed that high levels of social support from family were a main protective factor against suicide attempts in socially disadvantaged groups. In addition to family support, the concern for the potential impact on their families was identified by men in our study, which prevented suicide. Other scholars have also reported on this protective effect. For example, Shand et al. [25] found that 67% of men in their survey endorsed the item “I thought about the consequences for my family”, which offered some protection against men’s suicide. Social support and connectedness extend beyond the family [26]. Having local social support structures and an increased sense of community have been identified as protective factors against psychological distress in rural men [27]. With respect to relational connectedness, our findings indicate that the presence of significant others—particularly children—served as a protective factor, as the men did not want to cause their families pain. Social connections and support have been widely documented in the literature. 

Participants found meaning in exercise, being in nature, meditating, a positive workplace, pet ownership, and, for some, their faith. Traditional approaches to self-management of difficulties and struggles in daily life often concentrate on psychological means of behavior change, which have been shown to have some effectiveness in managing distress. These approaches seldom account for the wider resources in people’s surroundings that may hold significant relevance, such as meaningful connections with people, with nature, and with pet ownership. For example, the impact of dog ownership on physical activity has been previously reported with largely positive impacts on mental health and well-being [28].

Much of the current literature on suicide studies focuses on community empowerment and its positive role in suicide prevention [29]. In our study, men’s narratives focused instead on personal empowerment via a sense of inner strength and agency. Hope and optimism were expressed by a few men with the belief that a better life is both possible and attainable. The CHIME domain of identity, which encompasses the dimension of rebuilding or redefining a positive sense of identity and overcoming stigma, was the one CHIME domain that was not readily evident in the narratives of the men. This could have resulted from a lack of a strong sense of identity or from masculine norms of identity that posed a risk factor rather than a protective factor. A burgeoning body of research explores masculinity and gender roles as risk factors for suicidal ideation and suicide [30]. Some evidence shows males may be at increased risk of death by suicide as they are socialized to conform to certain masculine norms. Future research could further explore the concepts of masculinity and the intersection with relational connectedness—particularly intimate romantic relationships—for men in cultures of masculinity.

We acknowledge that it is difficult to know what men who take their own lives (on the first occasion) would consider protective effects. However, we would argue that this paper, which draws on the narratives of men, offers some insight into factors that may be protective for some men. Many who take their own lives/at risk of suicide do eventually go on to die from suicide, and the risk is higher; hence, this is likely the best method to get some insights. Often, death/survival depends on external circumstances, so it is likely that the same usefully reflects the perspectives of those who may not survive. While this research focuses on the protective factors that kept these men alive, our paper does not shed light on why other men choose to die by suicide [31].

Results indicate that psychosocial rehabilitation and holistic approaches targeting these protective factors—in particular, the uniquity of relational connections as a protective factor alongside its overlap with the other CHIME domains—may offer a promising avenue of exploration. Humanistic models have been suggested, which emphasize the importance of establishing meaningful social relationships, experiences, and participation in life—a process that entails significantly more than simply “treating” the “condition” of suicidality [32]. A complex, situated, and processual understanding of protective factors is needed. Moving away from an individual-focused perspective and facilitating community engagement in suicide prevention is critical. The World Health Organization [33] has called for such an active and participatory bottom-up approach where communities identify, prioritize, and implement activities that are important and appropriate to the social context. This would go a long way toward addressing the need for relational connections. We concur with the recent call for community-based approaches that are underpinned by positive psychological principles, such as community empowerment, wherein community needs and values are considered. A relatively new paradigm, critical suicidology, questions the status quo in understanding and preventing suicide. It offers a more subjective, historical, ecological, social justice-oriented, poetic, and socio-political approach than customary suicide studies. This more contextualized approach allows suicide prevention efforts to capitalize on community strengths. The pervasiveness of recreational connectedness aligns well with this goal of capitalizing on community strengths. 

Community-based participatory approaches highlight individual agency, cultural knowledge, and social change and are especially effective for use with marginalized groups. Participatory methods easily map on to the history of community-engaged responses to suicide and align well with socio-environmental difficulties that contribute to suicide, such as racial discrimination, bullying, poverty, and violence. Community intervention efforts should involve a collection of efforts to structure suicide prevention within the context of community history and needs and must involve and receive support from community members at all stages of implementation. By respecting and considering sociocultural determinants of health and what they mean to community members and by promoting community growth through suicide prevention, efforts to prevent suicide will be increasingly value-driven, motivated, and sustained. As such, this critical framework can support a positive psychological approach by providing a base for understanding the needs, strengths, and goals for growth of a community suicide prevention effort [34].

There is an increasing and pressing acknowledgement that different forms of knowledge about suicide are needed, along with more interdisciplinary ways of approaching its study [35]. Although psychiatric disorders significantly increase suicide risk, interpersonal and social factors also play an important role. A sociological analysis of suicide can be traced back to Durkheim [36] who argued that suicide results from the social context, which he refers to as social disorganization. Research investigating suicide from within sociology and political studies remains limited, and there continues to be little qualitative and interdisciplinary knowledge in this field [35]. Tang et al. [37] suggest that equity issues and broader policy reform in relation to social welfare, employment, education, and housing should be considered when exploring the relationship between social determinants of mental health and mental health service use among people who die by suicide. In addition, while protective factors in general might be universally applicable globally, our findings may need to be extended/replicated in other countries.

Understanding the subjective and lived experiences of those affected by suicidality has important potential to inform the innovation of culturally appropriate health services, shape public policy, and promote meaningful social change [38].

## 5. Conclusions

The importance of relational connections for men who have experienced suicidal thoughts or behaviors cannot be understated. These connections were highlighted in the deeply personal narratives of these men, who indicated that connections to family and peers, pets, and even nature were important protective factors in their lives that served to prevent suicide. 

## Figures and Tables

**Figure 1 ijerph-20-02259-f001:**
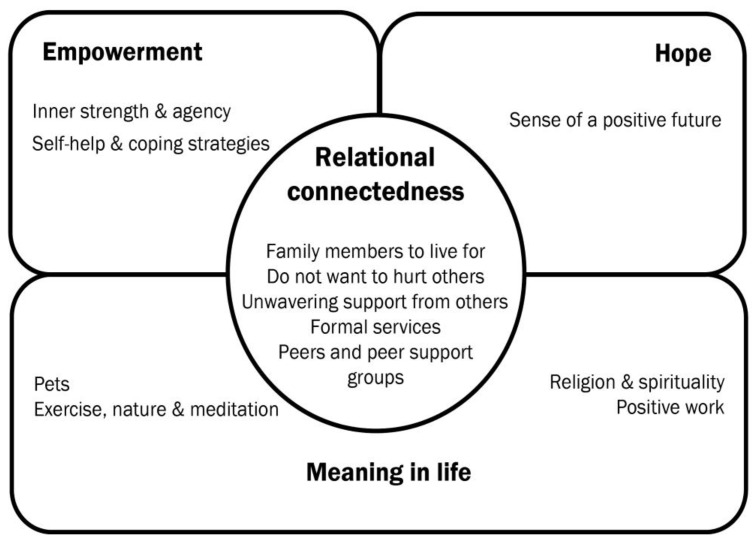
Protective factors mapped to the CHIME framework.

**Table 1 ijerph-20-02259-t001:** Participant demographics (n = 37 men).

Mean age (years)	47.5 R = 18–68
Ethnicity (%)	Caucasian = 35 (95) Asian = 1 (2.5) Aboriginal Torres Strait Islander = 1 (2.5)
Marital status	Married/de facto 18 (49) Single = 10 (27) Separated/divorced = 7 (19) Widowed = 1 (2.5) Unknown = 1 (2.5)
Sexuality	Heterosexual 32 (86) Homosexual = 3 (8.1) Asexual = 1 (2.7) Questioning = 1 (2.7)
Living status	With others = 27 (73) Alone = 9 (24) Unstable = 6 (16.2)
Education	Graduate/undergraduate = 20 (54) High school or less 17 (46)
Employment status	Employed (full-time) = 15 (41) Employed (part-time) = 4 (11) Unemployed (due to disability) = 10 (27) Retired = 3 (8) Student = 2 (5) Unemployed (not looking for work) = 1 (2.5) Casual work = (2) (5)
Rural/urban	Inner city = 33 (89) Regional/rural = 4 (11)

## Data Availability

The data are not publicly available due to confidentiality and anonymity issues.

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
