# Peer review of "Understanding Protective Factors for Men at Risk of Suicide Using the CHIME Framework: The Primacy of Relational Connectedness"

_ijerph, 2023, doi:10.3390/ijerph20032259_

Round 1
Reviewer 1 Report
Overall, the manuscript is well-written and organized. Given the authors purpose of the study there have been no similar research known to them about protective factors for "under-the-radar" men. I have no major concerns with the manuscript. I would have like to see the interview questions, though even if provided in an Appendix.
Author Response
Dear Reviewer 1: Thank you for your kind comments regarding our manuscript. As per your suggestion, we have added the interview questions in an Appendix.
Reviewer 2 Report
Thank you for the opportunity to review this important paper “Understanding Protective Factors for Men at Risk of Suicide using the CHIME Framework: The Primacy of Relational Connectedness”
I really enjoyed reading this paper - overall very well written, novel and really interesting. I only have a few suggestions for the authors to address:
The novel application of the CHIME model to suicide is great and shows it has universal application.
Abstract
The claim that most people who die by the suicide are under the radar- well I agree but maybe add a caveat that this is a critical suicidology view and that there is literature from psychological autopsy studies that is makes the opposite claim – that suicide is most often linked to mental illness - and maybe to be balanced you state thee are views/evidence both ways? Maybe the evidence is stronger in one direction?.
On the point that – “The pervasiveness of relational connectedness in men’s narratives was identified as a central protective factor is interesting and consistent with Fincham and Scourfields finding is his work on sociological autopsies. Understanding Suicide: A Sociological Autopsy (B. Fincham, S. Langer, J. Scourfield, M. Shiner) and Sociological autopsy: An integrated approach to the study of suicide in men study of suicide in men Jonathan Scourfield, Ben Fincham, Susanne Langer and Michael Shiner Published in Social Science & Medicine 74 (2012), pp. 466-473
I think these important works and their findings should feature in your introduction and discussion.
In the abstract also - you don’t comment on the ‘I’ missing in CHIME framework – about identity – in the abstract – which was clearly less apparent in your data – which is very critical and interesting as (lack or or unaccepted) identity is a critical issue in many suicides especially LGBTQI - where family and others don't accept the sexual/gender orientation. It also relates to anomie/alienation suicide theories Durkheim etc - as do strong religious faiths - especially those where entry into the afterlife is believed to be affected by taking ones own life). You could add this to your discussion.
Also I think you need to note the limitations of the deductive method in you discussion - that typically when you go looking for specific things in data you tend to find them! There is a confirmation bias there.
The themes fit the data mostly although as you note the Identify part was often absent. You dont make as much of this as you should. You could also explore more and concepts of masculinity and the significance of relationships - especially intimate romantic ones, for men in cultures of masculinity.
I wonder what you would find if you did a case control study using CHIMES, or another study applying CHIME to a population of men who had never been suicidal? This might be worth suggesting under further research.
Also so would using CHIME to examine suicides by women and other genders/sexual orientations. Also note paper by Jenkin et all in thei modification of the Chime framework adding an ‘S’ for safety. Jenkin G, McIntosh J, Every-Palmer S. 2021. Fit for what purpose? Exploring bicultural frameworks for the architectural design of acute mental health facilities IJERPH. https://www.mdpi.com/1660-4601/18/5/2343
Maybe you will find that lack of strong identity and acceptance of that identity is common amongst suicidal people. I suspect this is definitely the case - I think you could discuss this.
Finally. I think some of the info in the discussion might be better in the introduction. Especially the bits about critical suicidology - to prepare the reader for you approach and angle and findings.
Thank you again for a great paper - I hope these suggestions are helpful.
Author Response
Dear Reviewer #2:
Thank you for your thoughtful review of our paper. Our responses are highlighted in purple font below.
Thank you for the opportunity to review this important paper “Understanding Protective Factors for Men at Risk of Suicide using the CHIME Framework: The Primacy of Relational Connectedness”
I really enjoyed reading this paper - overall very well written, novel and really interesting. I only have a few suggestions for the authors to address:
The novel application of the CHIME model to suicide is great and shows it has universal application.
Abstract
The claim that most people who die by the suicide are under the radar- well I agree but maybe add a caveat that this is a critical suicidology view and that there is literature from psychological autopsy studies that is makes the opposite claim – that suicide is most often linked to mental illness - and maybe to be balanced you state there are views/evidence both ways? Maybe the evidence is stronger in one direction?
This has now been noted in the manuscript.
On the point that – “The pervasiveness of relational connectedness in men’s narratives was identified as a central protective factor is interesting and consistent with Fincham and Scourfields finding is his work on sociological autopsies. Understanding Suicide: A Sociological Autopsy (B. Fincham, S. Langer, J. Scourfield, M. Shiner) and Sociological autopsy: An integrated approach to the study of suicide in men study of suicide in men Jonathan Scourfield, Ben Fincham, Susanne Langer and Michael Shiner Published in Social Science & Medicine 74 (2012), pp. 466-473
I think these important works and their findings should feature in your introduction and discussion.
Thank you for highlighting these very interested works – we have added them to the paper as suggested.
In the abstract also - you don’t comment on the ‘I’ missing in CHIME framework – about identity – in the abstract – which was clearly less apparent in your data – which is very critical and interesting as (lack or or unaccepted) identity is a critical issue in many suicides especially LGBTQI - where family and others don't accept the sexual/gender orientation. It also relates to anomie/alienation suicide theories Durkheim etc - as do strong religious faiths - especially those where entry into the afterlife is believed to be affected by taking ones own life). You could add this to your discussion.
We have added the ‘I’ to the abstract as well as comments on the identity component in the discussion section.
Also I think you need to note the limitations of the deductive method in you discussion - that typically when you go looking for specific things in data you tend to find them! There is a confirmation bias there.
Thank you for pointing this out. We have noted our deductive method and its limitations as suggested.
The themes fit the data mostly although as you note the Identify part was often absent. You don’t make as much of this as you should. You could also explore more and concepts of masculinity and the significance of relationships - especially intimate romantic ones, for men in cultures of masculinity.
We have added to the discussion the absence of the identity theme in the interview narratives.
I wonder what you would find if you did a case control study using CHIMES, or another study applying CHIME to a population of men who had never been suicidal? This might be worth suggesting under further research.
We have added this suggestion to further research.
Also so would using CHIME to examine suicides by women and other genders/sexual orientations. Also note paper by Jenkin et all in their modification of the Chime framework adding an ‘S’ for safety. Jenkin G, McIntosh J, Every-Palmer S. 2021. Fit for what purpose? Exploring bicultural frameworks for the architectural design of acute mental health facilities IJERPH. https://www.mdpi.com/1660-4601/18/5/2343
We have included this very interesting paper. Jenkin GLS, McIntosh J, Every-Palmer S. Fit for What Purpose? Exploring Bicultural Frameworks for the Architectural Design of Acute Mental Health Facilities. International Journal of Environmental Research and Public Health. 2021; 18(5):2343. https://doi.org/10.3390/ijerph18052343
Maybe you will find that lack of strong identity and acceptance of that identity is common amongst suicidal people. I suspect this is definitely the case - I think you could discuss this.
This has been added to the discussion section.
Finally. I think some of the info in the discussion might be better in the introduction. Especially the bits about critical suicidology - to prepare the reader for you approach and angle and findings.
We have signposted critical suicidology study material into the introduction section.
Thank you again for a great paper - I hope these suggestions are helpful.
Thank you for your thoughtful review!
Reviewer 3 Report
This small qualitative study attempts to look at the protective factors for men which is helpful and needed in the field of suicide prevention as the authors state in the beginning. The authors did not include the mental health status (diagnosis) of the participants which would be helpful to know. It's unfortunate many admitted in the interviews access to care in the last year when they had not at the outset of their participation, this makes me question their validity of acceptance to the project. While protective factors in general might be universally applicable globally, some might not be and this study might be best applicable for Australian population so it's generalizability is unclear to me. While qualitative in design, the authors could have helped the reader by giving some more quantitative results for each domain they looked at. I think the biggest challenge for me in this paper is the usefulness of it. For example, I know countless people who have died by suicide that leave children and spouses behind, pets as well, that have jobs and reasons to live, but they still took their life. So in many ways that these males reported those things kept them alive does not help the field understand why the others didn't survive.
Author Response
Dear Reviewer #3
Thank you for taking the time to provide a review of our manuscript. We greatly appreciate your comments and have addressed them below [in purple font].
This small qualitative study attempts to look at the protective factors for men which is helpful and needed in the field of suicide prevention as the authors state in the beginning. The authors did not include the mental health status (diagnosis) of the participants which would be helpful to know.
Thank you for acknowledging the importance of addressing the little-known data on protective factors for men.
With respect to the diagnosis of men in our study, we did not assume that mental ill-health was present – in fact, many of the men in our study explicitly talked about the fact that it was not mental illness that underpinned their suicidal thoughts or behaviours, but rather other social determinants such as relationships, financial status, workplace situation, and previous childhood trauma. Thus, we do not have the data regarding absence of mental ill health and are not sure how this would play out or be helpful.
It's unfortunate many admitted in the interviews access to care in the last year when they had not at the outset of their participation, this makes me question their validity of acceptance to the project.
Participants did have access to care – we found it very difficult to capture ‘pure’ under the radar. What is critical to our team is the fact that the men responding perceived themselves as being ‘not in care’.
While protective factors in general might be universally applicable globally, some might not be and this study might be best applicable for Australian population so it's generalizability is unclear to me. While qualitative in design, the authors could have helped the reader by giving some more quantitative results for each domain they looked at.
We agree that the study has a relatively small sample size, and that the findings may need to be extended/replicated in other countries. We have made note of this in the limitations section of the paper.
I think the biggest challenge for me in this paper is the usefulness of it. For example, I know countless people who have died by suicide that leave children and spouses behind, pets as well, that have jobs and reasons to live, but they still took their life. So in many ways that these males reported those things kept them alive does not help the field understand why the others didn't survive.
We agree with this comment - that it is pretty much impossible to know what men who take their own lives (on first occasion) would consider protective effects. However, we would argue that this does not make the paper useless.
- Many who take their own lives/at risk of suicide do eventually go on to die from suicide – the risk is higher; hence this is likely the best method to get some insights
- Often death/survival depends on external circumstances – so likely that the same usefully reflects the perspectives of those who may not survive
- While this research focuses on the protective factors that kept these men alive, our paper does not shed light on why other men choose to die by suicide (for this, see Macdonald et al., 2022).
We have made note of this in the limitations section